# Preparation of Complementary Food for Infants and Young Children with Beef Liver: Process Optimization and Storage Quality

**DOI:** 10.3390/foods12142689

**Published:** 2023-07-13

**Authors:** Ruheng Shen, Dawei Yang, Li Zhang, Qunli Yu, Xiaotong Ma, Guoyuan Ma, Zhaobin Guo, Cheng Chen

**Affiliations:** College of Food Science and Engineering, Gansu Agriculture University, Lanzhou 730070, China; ruhengshen@163.com (R.S.); ydw980414@163.com (D.Y.); yuqunlihl@163.com (Q.Y.); maxiaotong17@163.com (X.M.); maguoyuangsau@163.com (G.M.); guozhb007@163.com (Z.G.); chenchengmlj@163.com (C.C.)

**Keywords:** infant supplement, beef liver, fuzzy mathematics, nutrition, digestibility

## Abstract

In this study, fuzzy mathematics and response surface modeling were applied to optimize the preparation process of beef liver paste and characterize the proximate composition, sensory and physicochemical qualities, and in vitro simulated digestive properties while refrigerated at 0–4 °C (0, 3, 7, 15, 30, 45, and 60 days). The results showed that the optimal preparation process was 4.8% potato starch, 99.4% water, 10.2% olive oil, and a 3:2 ratio of chicken breast and beef liver. The beef liver paste prepared contained essential amino acids for infants and children, with a protein content of 10.29 g/100 g. During storage, the pH of the beef liver paste decreased significantly (*p* < 0.05) on day 7, texture and rheological properties decreased significantly after 30 days, *a** values increased, *L** and *b** values gradually decreased, and TVB-N and TBARS values increased significantly (*p* < 0.05) on day 7 but were below the limit values during the storage period (TVB-N value ≤ 15 mg/100 g, TBARS value ≤ 1 mg/Kg). In vitro simulated digestion tests showed better digestibility and digestive characteristics in the first 15 days. The results of this study provide a reference for the development of beef liver products for infant and child supplementation.

## 1. Introduction

The first two years of infant and toddler life are critical periods of rapid growth and development [1]. The Global Strategy for Infant and Young Child Feeding (2003) recommends the need to provide complementary feeding for infants and young children from 6 months of age, an intervention that can have long-term effects on growth and development, even into adulthood. Complementary foods for infants and young children are foods provided for the gradual adaptation of weaned infants to conventional foods or dietary supplements for young children (Regulation (EU) 609/2013). Complementary foods for infants and young children are of plant and animal origin, with proteins from animal foods widely considered to have higher nutritional quality than plant proteins [2]. Animal liver can be used as a supplementary food, and beef liver is one of the sources [3] and contains high-quality protein and is rich in trace elements such as heme iron, zinc, and B vitamins, which are important nutrients for cognitive function, neurological function, and brain growth and development in infants and children [4,5]. In particular, heme iron is readily absorbed by infants and children and enhances the absorption of non-heme iron from foods such as cereals, vegetables, and legumes [6,7].

Animal livers can be processed into foods with good organoleptic properties, such as liver paste, liver powder, and liver sausage [8], and are popular worldwide [9]. Currently, the United States [10], Germany [11], and Australia [12] encourage meat and meat products (chicken, pork, beef, animal liver, etc.) as an important source of complementary foods for infants and young children. In a study of Peruvian infants and children, chicken legs and livers were cooked, then freeze-dried and ground into a fine powder and added to porridge as a supplemental source of iron and zinc for infants and children [13]. Duizer et al. [14] subjected beef and beef livers to hot water steaming, two-step hot oil treatment, and freeze-dried in a series of processing steps and finally ground into powder as a fortifying supplement added to food. A series of studies was also conducted on 96 Indonesian infants and toddlers aged 12 to 23 months. The results showed that the micronutrients required were obtained with the addition of meat meal and that infants and young children readily accepted foods containing meat meal. Amaral et al. [15] prepared a lamb liver paste containing 12% lamb meat, 25% lamb liver, 13% lamb blood, 20% water, and 30% fat. The product is of high nutritional value, the iron content is 9.0 mg/100 g, and it also contains essential linoleic acid.

Since infants and toddlers’ chewing ability is not fully developed, food is usually processed into purees to help them transition from liquid foods to semi-solid or solid foods [16]. In addition, during meat processing, some form of starch is usually added to improve the texture [17]. Starch is a major source of dietary glucose required for the rapid development of infants and young children and can be added to special formulations of infant foods or supplements [18]. Vegetable oils rich in polyunsaturated fatty acids are also added, which can improve the fatty acid composition [19] and act as a fortification vehicle for supplemental feeding. Pork liver paste prepared by Rezler et al. [20] showed reduced firmness, spreadability, and adhesion with the addition of modified starch and improved palatability due to reduced fat content. Bilska et al. [21] added 20% flaxseed oil to pork liver paste, which resulted in a reduction in saturated fatty acids and monoenoic acids in the product by about 12% and increased the content of polyenoic fatty acids by more than 70%, which facilitated the reduction in cholesterol content in terms of nutrition.

It has been reported that because the gastrointestinal tract of infants and young children is not fully developed, protein digestibility in the gastrointestinal tract is low, reducing protein availability [22,23]. Moreover, meat products are also susceptible to protein oxidation and lipid oxidation during storage, leading to a decrease in quality and nutritional value [24], which can also affect protein digestibility. There is a lack of adequate reporting on studies of changes in product quality during storage affecting digestibility and digestive characteristics. The results of this study provide new ideas for the development of beef liver products for use in complementary foods for infants and young children, which could enrich the variety of complementary foods while increasing the added value of bovine by-products.

## 2. Materials and Methods

### 2.1. Materials

Beef liver provided by Gansu Kangmei Modern Agriculture and Animal Husbandry Industry Group Co., Linxia, China (three 1.5-year-old bulls with good growth and development; their average weight was approximately 300 kg; all 3 originated from the same feeding lot, and were of similar body condition, healthy, and disease-free). The beef liver was washed of blood and impurities; removed from connective tissue, sinew, and fat; and cut into strips of 3 cm × 3 cm × 5 cm. Chicken breast, olive oil, potato starch, and salt were purchased at the local supermarket. All chemicals and reagents used were analytical grade.

### 2.2. Optimization of Preparation Technology of Beef Liver Paste as a Complementary Food for Infants and Young Children

#### 2.2.1. Preparation of Beef Liver Paste

Figure 1 depicts the experimental design for the preparation of beef liver paste as a supplement for infants and children. Deodorization of the beef liver: The beef liver was placed in a certain amount of room temperature water with 1% salt by weight of beef liver and soaked at 4 °C for 1 h. The beef liver and chicken breast were boiled in water at 95–100 °C for 15 min and put in a chopper; water was added, and they were chopped for 1–3 min until they were paste-like. Potato starch and olive oil were added to chopped meat and chopping was continued for 1 min. The chopped meat was canned and autoclaved for 15 min at 121 °C. The cooled beef liver paste was stored at 0–4 °C.

#### 2.2.2. Single-Factor Experiment

All percentages were expressed by weight (*w/w*). Four single-factor experiments were performed, including potato starch addition (2%, 4%, 6%, 8%, 10%), water (50%, 70%, 90%, 110%, 130%), olive oil (2%, 6%, 10%, 14%, 18%), and chicken breast to beef liver ratio (7:3, 3:2, 1:1, 2:3, 3:7). The single-factor test was conducted with texture and fuzzy mathematical sensory score as evaluation indicators.

#### 2.2.3. Response Surface Methodology

A 4-factor, 3-level Box–Behnken design [25] was used to investigate the effects of potato starch, water, olive oil, chicken breast, and beef liver on the hardness, gumminess, and sensory scores of beef liver paste. The weight values for hardness, gumminess, and fuzzy mathematical sensory scores of beef liver paste were set at 30%, 30%, and 40%, respectively.

#### 2.2.4. Fuzzy Modeling

A fuzzy mathematical sensory evaluation method was used to comprehensively evaluate the color, aroma, taste, and tissue condition of beef liver paste for infant and toddler complementary foods. Ten trained and qualified evaluators were selected to randomly evaluate the beef liver paste and ranked according to the characteristics of the paste from “excellent” to “poor” quality.

The set of factors was determined according to Table 1. The first-order factor set U = [U_1_ U_2_ U_3_ U_4_], wherein U_1_, U_2_, U_3_, and U_4_ represent taste, tissue state, flavor, and color, respectively. The secondary factor is set U_1_ = [U_11_ U_12_ U_13_], where U_11_, U_12_, and U_13_ represent the fineness, softness, hardness, and residue amount, respectively. The secondary factor set U_2_ = [U_21_ U_22_], where U_21_ and U_22_ represent the water separating property and the smearing property, respectively. The “0, 4” scoring method was used to determine the weight of each factor. The evaluation resulted in four levels of evaluation. The evaluation set was V = [V_1_, V_2_, V_3_, V_4_], wherein V_1_, V_2_, V_3_ and V_4_ represented excellent, good, medium, and poor.

Comprehensive fuzzy relation evaluation set: The members of each evaluation group evaluated each factor, counted the votes of each factor evaluation grade, and obtained the corresponding fuzzy matrix of each factor. Grading upward by the weight of the last factor was performed. The comprehensive evaluation set was obtained according to the fuzzy matrix, and the calculation formula was shown in the following equation:(1)K=C×F
where: *C* represents the weight set and *F* represents the fuzzy relation matrix.

#### 2.2.5. Texture Profile Analysis

Referring to Terrasa et al. [26], a texture analyzer (TA.XT Express, Stable Micro Systems, Beijing, China) was used to measure the texture profile of the beef liver paste. The measurement parameters were: 5 mm diameter probe and pre-test speed, test speed, and post-test speed of 2 mm/s, 2 mm/s, and 5 mm/s, respectively, with a force of 5 g. Six sets of data were measured for each sample group.

#### 2.2.6. Determination of Nutritional Indicators of Beef Liver Paste

##### Chemical Composition

Moisture, fat, ash, and protein content were determined by standardized methods developed by the Association of Official Analytical Chemists (AOAC), Rockville, MD, USA. The sample was dried to constant weight in an oven at 105 °C to determine the moisture content. Fat was determined by Soxhlet extraction method. The ash content was measured by carbonizing the sample in a high-temperature furnace at 500–600 °C with a crucible. The protein content was determined by Kjeldahl method.

##### Amino Acid Determination

According to the modified method of Hamed Hammad Mohammed et al. [27], 25 mg of lyophilized sample was taken into a sample vial, and 10 mL of 6 mol/L hydrochloric acid was added; the vial was sealed and then hydrolyzed it in an oven at 110 °C for 24 h. After completion of hydrolysis, the volume was fixed to 10 mL, and 1 mL of filtrate was taken and dried in an oven for 1 h. After completed evaporation, the solution was dissolved with 3 mL of distilled water and filtered, and 1 mL was taken into a sample injection bottle and analyzed in an automated amino acid analyzer (L-8900; Hitachi, Tokyo, Japan).

##### Fatty Acids

According to the modified method of Xiong et al. [28], the fatty acid profile of beef liver paste was analyzed by gas chromatography (GC-6850, Agilent, Santa Clara, CA, USA). The nitrogen flow rate was set at 1.2 mL/min and the air flow rate was set at 450 mL/min. The column was isothermally operated at 5 °C/min at 140–240 °C and held at 240 °C for 15 min. The injection temperature and detector temperature were 260 °C and 250 °C, respectively. Hydrogen (40 mL/min) was used as the carrier gas. The isolated fatty acids were identified by comparison with the retention time of the standard solution. The results are reported in grams of fatty acids per 100 g of beef liver paste.

### 2.3. Quality Indicators for Storage of Beef Liver Paste

#### 2.3.1. pH Value Measurement

The beef liver paste was ground in a wall breaker, weighed to 10 g, and then distilled water was added, and the mixture was homogenized and mixed. The pH was determined using a portable pH meter (Testo 205 portable waterproof pH; Testo Instruments International Trading Co., Ltd., Shanghai, China) according to the method of Sogut and Seydim [29]. Three replicates were performed for each sample.

#### 2.3.2. Color Measurement

Measurements were made using a colorimeter CR-10 (Beijing Comerun Instruments Co., Ltd., Beijing, China) according to the method of Sogut and Seydim [29]. The colorimeter was calibrated according to black and white standards. *L** (lightness), *a** (redness), and *b** (yellowness) measurements were taken and recorded as the average of these measurements. The total color difference (∆E) was calculated by the following equation:(2)$$\Delta E=\sqrt{\left[{\left(L^{*}-L_{0}^{*}\right)}^{2}+{\left(a^{*}-a_{0}^{*}\right)}^{2}+{\left(b^{*}-b_{0}^{*}\right)}^{2}\right]}$$

*L**, *a** and *b** values at day 0 were chosen as reference color values ($L_{0}^{*}$, $a_{0}^{*}$, $b_{0}^{*}$). Each sample was measured three times.

#### 2.3.3. Rheological Measurement

Huang et al. [30] used a DHR-1 rheometer (TA Instruments, New Castle, DE, USA) to determine the rheological properties of beef liver paste. The beef liver paste was placed between two parallel plates with a fixed gap of 1 mm, and the apparent viscosity was measured at shear rates of 0.01–100 s^−1^. Frequency scan: the measurements were performed at dynamic frequencies of 0.1–100 Hz to determine the elasticity modulus G’, loss modulus G’’, and loss tangent tan δ of the beef liver paste.

#### 2.3.4. TVB-N Measurement

TVB-N was determined by the method described by Cheng et al. [31]; beef liver paste (10 g) was dispersed in 75 mL distilled water and equilibrated for 30 min. Then it was poured into a distillation tube and 1g magnesium oxide was added and distilled by Kjeldahl nitrogen analyzer, and the distillate was mixed with 20 g/L boric acid solution. TVB-N was determined by hydrochloric acid and was calculated using the following formula:(3)$$\mathrm{TVB-N}\left(\mathrm{mg}/100g\right)=\left(V_{1}-V_{2}\right)\times c\times 14/m\times 100$$
where *V*_1_ is the volume of standard hydrochloric acid titration solution consumed by the sample (mL); *V*_2_ is the volume of standard hydrochloric acid titration solution consumed by the blank reagent (mL); *c* is the concentration of standard hydrochloric acid titration solution (mol/L); m is the sample mass (g); 14 is the mass of nitrogen equivalent to titrate 1.0 mL of hydrochloric acid [*c* (HCl) = 1.000 mol/L] standard titration solution (g/mol).

#### 2.3.5. TBARS Measurement

TBARS was determined as described by Zhang et al. [32]. Briefly, 10.0 g of beef liver paste was placed in a beaker, and 50 mL of 7.5% trichloroacetic acid solution (containing 0.1% EDTA) was added, stirred well with a glass rod, and filtered; 5 mL of supernatant was added, and 5 mL of 0.02 mol/L thiobarbituric acid solution was added, heated in a water bath at 90 °C for 45 min, removed and cooled, and centrifuged at 4500 r/min for 10 min. Absorption values of the supernatant were measured at 532 nm and 600 nm, and TBARS values were calculated. Results were expressed as mg malondialdehyde (MDA) per kilogram of beef liver paste.

#### 2.3.6. In Vitro Simulation of Digestion

##### Digestibility

Simulated gastric fluid (SGF) and simulated intestinal fluid (SIF) were prepared according to Luo et al. [33]. Briefly, 2 g of sample was weighed, 8 mL of SGF was added, and the reaction was carried out in a water bath at 37 °C for 2 h. After the reaction, the pH of the enzyme hydrolysate was adjusted to 7.0 with 0.1 mol/L NaOH solution to form the gastric enzyme digest. To the pepsin digest, 0.1 mL SIF was added, and the pH of the digest was adjusted to 7.5 with 0.1 mol/L NaOH, and the digest was water-bathed at 37 °C for 2 h. After the water bath, the digest was heated at 95 °C for 5 min and then cooled. The cooled mixture was a two-step enzymatic hydrolysate of the stomach and gastrointestinal tract.

The gastric hydrolysis product and gastrointestinal hydrolysis product were added to 20 mL of anhydrous ethanol and centrifuged (4500 r/min, 4 °C) for 10 min. After centrifugation, the supernatant was discarded and the precipitate was dried at 50 °C to a constant weight, and the protein content of the samples before and after digestion was determined by Kjeldahl nitrogen determination. The digestion rate was calculated as follows:(4)$$DT\left(\%\right)=\left(W_{0}-W_{1}\right)/W_{0}\times 100$$
where *DT* indicates the protein digestibility of beef liver paste for infant supplement, *W*_0_ indicates the protein content in the sample before digestion (g), and *W*_1_ indicates the protein content in the sample after drying (g).

##### Particle Size

A Mastersizer 3000 laser particle size analyzer (Malvern Instruments, Malvern, UK) was applied to measure the particle size of beef liver paste [33]. Parameters: the dispersion medium was water, the refractive index of the dispersed phase was 1.33, the shading was between 8% and 20%, and the refractive index and absorbance of the samples were 1.54 and 0.001.

##### Sodium Dodecyl Sulfate-Polyacrylamide Gel Electrophoresis (SDS-PAGE)

According to the method of Lee et al. [34], polyacrylamide gel electrophoresis denaturing experiments were performed using 12% separated gels and 4% concentrated gels. The sample was diluted and mixed with the upper sample buffer to a concentration of 7 mg/mL. All samples were heated at 100 °C for 5 min and then loaded onto the gel with a loading volume of 15 μL. The gel was run at a voltage of 70 V and a current of 80 A. The gel was then heated for 5 min at 100 °C and then loaded onto the gel with a loading volume of 15 μL. After completion of electrophoresis, the film was removed and stained for 2–3 h. The film was then decolorized with a decolorizing solution, and the strips were clear and photographed with the gel imaging system.

### 2.4. Statistical Analysis

Data were processed (mean and standard deviation calculated) using Excel 2019 and analyzed for significance (*p* < 0.05) using SPSS 20.0. Plotting, correlation analysis, principal component analysis, and cluster analysis were performed using Origin (version 2021).

## 3. Results and Discussion

### 3.1. Modeling and Optimization by Fuzzy Model

#### 3.1.1. Sensory Evaluation Results of Fuzzy Mathematics

Ten members of the sensory evaluation team scored the sensory quality of beef liver paste. The specific scores are shown in Table 2. The weight of each primary factor can be obtained from the table, $C_{U}=\left[\begin{array}{cccc} 0.31 & 0.18 & 0.28 & 0.23 \end{array}\right]$, and the weight of each secondary factor is $C_{U1}=\left[\begin{array}{cccc} 0.31 & 0.18 & 0.28 & 0.23 \end{array}\right],$ $C_{U2}=\left[\begin{array}{cccc} 0.31 & 0.18 & 0.28 & 0.23 \end{array}\right]$.

Combined evaluation of each secondary factor for the beef liver paste with serial number 1:$$K_{U1}{=C}_{U1}{\times F}_{U1}=\left[\begin{array}{ccc} 0.44 & 0.34 & 0.22 \end{array}\right]\times \left[\begin{array}{cccc} 7 & 2 & 1 & 0\\ 6 & 3 & 1 & 0\\ 8 & 1 & 1 & 0 \end{array}\right]=\left[\begin{array}{cccc} 0.688 & 0.212 & 0.1 & 0 \end{array}\right]\;K_{U1}{=C}_{U1}{\times F}_{U1}=\left[\begin{array}{cc} 0.4 & 0.6 \end{array}\right]\times \left[\begin{array}{cccc} 4 & 4 & 2 & 0\\ 3 & 3 & 0 & 0 \end{array}\right]=\left[\begin{array}{cccc} 0.58 & 0.37 & 0.08 & 0 \end{array}\right]$$

The fuzzy matrix of each level of factors can be obtained:$$F={\left[\begin{array}{cccc} K_{U1} & K_{U2} & K_{U3} & K_{U4} \end{array}\right]}^{T}$$

And the weights of each factor set are:$$C=\left[\begin{array}{cccc} 0.31 & 0.18 & 0.28 & 0.23 \end{array}\right]$$

Therefore, the comprehensive sensory evaluation of sample No. 1 can be obtained as follows:$$K=C\times F=\left[\begin{array}{cccc} 0.31 & 0.18 & 0.28 & 0.23 \end{array}\right]\times \left[\begin{array}{cccc} 0.688 & 0.212 & 0.1 & 0\\ 0.58 & 0.37 & 0.08 & 0\\ 1 & 3 & 4 & 2\\ 3 & 4 & 3 & 0 \end{array}\right]$$

The fuzzy mathematical sensory score of sample number 1 was obtained at 71.91 by multiplying the comprehensive sensory evaluation of sample number 1 by the score set. Fuzzy mathematical sensory scores for other samples were obtained according to this calculation method.

#### 3.1.2. Single-Factor Experimental Results

##### Effect of Adding Potato Starch to Beef Liver Paste on Texture and Sensory Evaluation

The addition of starch maintains the desired appearance and texture of foods [35]. From Figure 2a, it can be seen that with the increase in potato starch addition, the hardness and gumminess of beef liver paste increased significantly (*p* < 0.05), and the sensory score showed a trend of ascending and then descending, reaching the maximum value of potato starch addition of 6%. The paste tissue was relatively loose when the amount of potato starch was low. And when the amount of addition gradually increased, the beef liver paste became softer, smoother, and more delicate. This is due to the mixing of starch with saliva, which leads to faster digestion of starch, and the fact that α-amylase protein breaks the glycosidic bonds in starch, resulting in a decrease in product viscosity [36]. In agreement with the results of Varga-Visi et al. [37], the hardness and chewiness of turkey sausage were proportional to the amount of potato starch added. However, when there is too much potato starch, the paste tissue is dry and distinctly grainy, which is not conducive to swallowing by infants and children and may be a choking hazard [38]. Therefore, 4%, 6%, and 8% potato starch were selected for response surface optimization.

##### Effect of Adding Water to Beef Liver Paste on Texture and Sensory Evaluation

The addition of water to beef liver paste ensures that the paste body is moist and adheres [36]. Figure 2b shows that the hardness and gumminess of beef liver paste decreased significantly (*p* < 0.05) with the gradual increase in water addition. When the water content reached 90%, the beef liver paste had a homogeneous texture, a fine taste, and the highest overall acceptability. This may be due to the increased thickness of the beef liver paste, resulting in increased oral dwell time and mixing intensity, improving lubrication during swallowing [39]. Therefore, 70%, 90%, and 110% of water were selected for response surface optimization.

##### Effect of Adding Olive Oil to Beef Liver Paste on Texture and Sensory Evaluation

The addition of olive oil improves the viscoelasticity of beef liver paste [40], and olive oil is rich in DHA, which increases the content of increased unsaturated fatty acids and changes its fatty acid structure [41]. According to the results in Figure 2c, the hardness and gumminess of beef liver paste decreased significantly (*p* < 0.05) with the addition of olive oil. This is due to the emulsifying effect of olive oil, which improved the stability of the beef liver paste and made its texture softer and more delicate. Similar to the results of Morales-Irigoyen et al. [42], the hardness of pork liver paste gradually decreased when the amount of pre-emulsified oil added was gradually increased. When added at 10%, the highest overall acceptance of sensory evaluation was observed. However, when too much olive oil was added, the quality of beef liver paste decreased due to the separation of water and oil from the paste. Therefore, 6%, 10%, and 14% olive oil contents were selected for response surface optimization.

##### Effect of Adding Ratio of Chicken Breast and Beef Liver on Texture and Sensory Evaluation of Beef Liver Paste

Chicken breast has finer and softer muscle fibers, which can reduce the hardness of meat products [43]. According to the results in Figure 2d, as the ratio of chicken breast increased, the hardness and gumminess of beef liver paste also increased gradually, and the sensory score increased significantly (*p* < 0.05). And the overall sensory acceptability of beef liver paste was highest when the ratio was 1:1. When the chicken breast ratio was too low, it had the fishy smell of beef liver. However, when the percentage of chicken breast was too high, the hardness of the beef liver paste increased, chewiness decreased, and sensory scores decreased (*p* < 0.05). Consistent with the findings of Kim et al. [44], the addition of chicken breast to mackerel sausage resulted in a product that was superior to sausages without chicken breast in terms of smell, taste, firmness, chewiness, and overall preference. Therefore, the ratios of 2:3, 1:1, and 3:2 for chicken breast and beef liver were selected for response surface optimization.

#### 3.1.3. Optimization

##### Optimization Results and Analysis of Variance

The Box–Behnken design was used to optimize the response surface method experiment, and the results are shown in Table 3. A multiple regression fit analysis of the experiment was performed using Design Expert 10.0 software, and the regression equation was Y = 0.67 + 0.034A + 0.16 − 0.12C + 0.019D − 5.000E − 0.03AB − 0.018AC + 0.000AD + 0.013BC + 2.500E − 0.03BD − 5.000E − 0.03CD − 0.10A^2^ − 0.15B^2^ − 0.11C^2^ − 0.098D^2^. The ANOVA results for linear, interactive, and quadratic relationships between the effects of the respective variables on the response are given in Table 4. Among them, the effects of independent variables A, B, C, D, and interaction terms AC and BC on the composite value Y were highly significant (*p* < 0.01); the effects of AB and CD on the composite value Y were significant (*p* < 0.05), while the effects of AD and BD on the composite value Y were not significant (*p* > 0.05). The effects of A^2^, B^2^, C^2^, and D^2^ on the composite value Y in the model all reached a highly significant level (*p* < 0.01). Therefore, the effects of each factor on the combined Y value were in the order of moisture addition > potato starch addition > ratio of chicken breast to beef liver addition > olive oil addition. The correlation coefficient R^2^ was 0.9997, and the correction coefficient of the experimental model R^2^_adj_ was 0.9995, indicating that the data were reliable, and the model fit was good and could be used for theoretical prediction of hardness, gelatinization, and sensory scores of beef liver paste.

##### Interaction Analysis among Factors

As shown in Figure 3, the influence of the four factors on the total score and the interaction between the four factors are described. The steeper the surface of the response surface, the more significant the effect of the interaction between the two variables on the integrated value Y. The results showed significant interactions between chicken breast and beef liver addition ratio A and moisture addition B, chicken breast and beef liver addition ratio A and potato starch addition C, moisture addition B and potato starch addition C, and potato starch addition C and olive oil addition D, and the results were significant and consistent with the ANOVA results in Table 4.

##### Validation Tests

After the correction, the optimal process conditions were potato starch addition of 4.8%, water addition of 99.4%, olive oil addition of 10.2%, and chicken breast and beef liver addition ratio of 3:2. Three validation tests were performed according to this condition: a hardness of 10.96 N, a gelling property of 7.23, a sensory score of 77.18. This result is close to the ideal model value (0.625), indicating that the process conditions for optimizing beef liver paste for infant supplementation using the response surface methodology are reliable.

### 3.2. Chemical Composition

Preliminary characterization of the chemical composition of beef liver paste (Table 5) indicated the moisture content of beef liver paste was 78.49 g/100 g, the total ash content was 1.13 g/100 g, the protein content was 10.29 g/100 g, and the fat content was 5.17 g/100 g. The content of chemical components was low in protein and high in fat compared to the liver paste reported by other authors [45,46]. The fat content of commercial meat paste detected by Ettinger et al. [47] was 3.3–13.6 g/100 g, and the protein content was 11.7–17.1 g/100 g. Compared to this, the fat content of beef liver paste was slightly lower, which may be due to the addition of potato starch and chicken breast to beef liver paste resulting in a lower fat content.

The amino acid composition of beef liver paste is shown in Table 5. 14 amino acids were detected in beef liver paste, including seven essential amino acids (isoleucine, leucine, threonine, phenylalanine, lysine, valine, histidine) and seven non-essential amino acids (aspartic acid, serine, glutamic acid, cysteine, methionine, tyrosine, arginine). Among them, glutamic acid had the highest content (3.19 g/100 g), followed by leucine and aspartic acid, and methionine had the lowest content. Essential amino acids play an important role in linear growth and neurocognitive development in infants and young children [48]. Among them, leucine is considered an effective nutritional signal for inducing muscle synthesis and bone density, and high-quality animal-based protein improves linear growth in infants and children at risk of growth retardation [49].

In the fatty acid composition of beef liver paste, oleic acid was the highest (412.98 mg/100 g), followed by linoleic acid, stearic acid, arachidonic acid, and α-linolenic acid. This was similar to the composition of olive oil, in which oleic acid has the highest content, followed by palmitic acid, linoleic acid, and stearic acid [50]. Beef liver paste contained 60.41% monounsaturated fatty acids (MUFA), 25.65% polyunsaturated fatty acids (PUFA), and 14.21% saturated fatty acids (SFA). Polyunsaturated fatty acids in pig liver products prepared by Estevez et al. [51] were 1.44 g/100 g and consisted mainly of omega-6 fatty acids (1.20%). This difference could be due to the addition of olive oil, which resulted in a higher content of polyunsaturated fatty acids. And supplementation with polyunsaturated fatty acids can promote infant brain development and improve attention span [52].

### 3.3. Analysis of Storage Quality of Beef Liver Paste

#### 3.3.1. Chemical Composition and Color

The changes in the chemical composition of beef liver paste during storage are shown in Table 6. The moisture content of beef liver paste decreased gradually at the beginning of storage, and by the 45th day of storage, the moisture content decreased significantly (*p* > 0.05), consistent with the findings of Mishra et al. [53]. Chevon Seekh Kabab showed insignificant decrease in moisture at 0–21 days of storage. The decrease in protein content at day 60 may be due to the loss of soluble protein as a result of water loss from the beef liver paste [54]. Ash content increased from 1.13 g/100 g to 1.18 g/100 g during storage. This change was similar to Salman et al. [55], where the moisture content of fish patties decreased from 61.16% to 54.3%, and the ash content increased from 3.4% to 4.42% after 16 days of refrigeration, probably due to a decrease in moisture content and an increase in ash content to obtain an approximate percentage balance of composition.

Color is one of the important parameters to reflect the freshness of meat products [56]. According to the results in Table 6, *L** values continued to decrease with increasing storage time, possibly due to evaporation of water from the surface of beef liver paste [57]. Moreover, lipid oxidation products produced during storage of beef liver paste can be used as reaction substrates to oxidize myofibrillar proteins and cause protein cross-linking, which affects the net charge of proteins and alters the structure and spatial arrangement of myofibrillar proteins, thereby reducing water retention [58], and the reduced water content gives beef liver paste a dull and shiny appearance. In the product processing, some natural antioxidant substances can be added to inhibit oxidation and ensure product quality. Zhang et al. [59] investigated the effect of cinnamon essential oil on the preservation of Italian-style sausages, which prolonged color shelf life by inhibiting lipid oxidation. Turan and Simsek [60] reported that black mulberry water extract (BMWE) reduced oxidation and discoloration of beef patties during storage, with significant inhibitory effects. Water loss also allowed the accumulation of myoglobin pigments on the surface of the paste [61], leading to an increase in *a** values (*p* < 0.05). The decrease in a* values may be due to the formation of myoglobin pigments on the surface of the product, where water loss allows the accumulation of myoglobin pigments on the surface of the sauce [59], resulting in the conversion of iron in the form of oxy-myoglobin ferrous to myoglobin ferric ions and an increase in *a** values (*p* < 0.05). The *b** value also decreased gradually with storage time because the reaction of lipid oxidation products with protein amino groups produces lipofuscin, resulting in changes in *b** [62]. In this study, ΔE was expressed as the difference between consecutive measurements, and the color difference was perceptible to the human eye when ΔE > 3 [63]. The results showed a significant color change in the first 15 days, with a very high perception of color change between 45 and 60 days as storage time increased (*p* < 0.05).

#### 3.3.2. pH and Sensory Score

The pH value reflects the acidity and alkalinity of the meat sample and is an important indicator of the freshness of the meat sample [64]. As shown in Figure 4a, the pH value of beef liver paste during storage ranges from 6.87 to 6.76. Statistical analysis showed that there was no significant difference in the pH of beef liver paste at the beginning of storage (*p* > 0.05), and after the seventh day, the pH of beef liver paste decreased significantly (*p* < 0.05), which may be due to the decrease in pH due to acid production by microorganisms during storage [65]. After 15 days, pH and sensory scores converged. After 30 days, the sensory scores of beef liver paste decreased significantly (*p* < 0.05) due to a decrease in pH, water leaching from the paste, and a decrease in water retention capacity, resulting in a dull gloss [66].

#### 3.3.3. Texture

The changes in textural properties of beef liver paste during storage are shown in Figure 4b. The springiness and resilience of beef liver paste did not change significantly (*p* > 0.05) during storage and were in a relatively stable state. The hardness of beef liver paste decreased significantly (*p* < 0.05) at day 15 and decreased further with longer storage time. Similar to the results of Zhang et al. [67], this was probably due to the hydrolysis of connective tissue and myogenic fibrin by enzymes secreted by microorganisms, resulting in a decrease in the hardness of beef liver paste. On day 7, cohesiveness and gumminess increased significantly (*p* < 0.05) and decreased significantly thereafter (*p* < 0.05). In the study of Ali et al. [68], the hardness and cohesiveness of tilapia lunch cans decreased by 22.86–47.69% and 40–41.33%. This was due to the decrease in water content during storage and the continuous breakdown of proteins resulting in a decrease in the cohesiveness of the product [69]. The high protein content of beef liver paste and the residual oxygen in the tank also accelerated protein degradation and the combined effect of microorganisms, which ultimately caused the reduction in the product’s structural properties.

#### 3.3.4. Rheological Properties

Infant paste is a semi-solid food that is easily swallowed by infants and young children due to its viscous state [33], and its texture is closely related to rheology. As shown in Figure 5a, the apparent viscosity decreased with increasing shear rate, indicating that beef liver paste is a non-Newtonian fluid with shear thinning properties. Moreover, at the beginning of storage, the apparent viscosity increased with storage time, reaching a maximum at day 7. This may be due to the oxidation of proteins in beef liver paste, which caused the weakening of cross-linking aggregation between proteins and the release of a large amount of moisture [70]. And the change in moisture content caused the apparent viscosity of beef liver paste to decrease in the later stages of storage. From day 30, the apparent viscosity decreased, possibly due to protein oxidation in the beef liver paste, resulting in weaker protein cross-linking and aggregation and the release of large amounts of water, resulting in a decrease in apparent viscosity.

G’ is the elasticity modulus that can be used to express the gel capacity of the protein [71]. G” is the loss modulus, which reflects the ability of the paste to resist flow. The elasticity modulus and loss modulus of beef liver paste gradually increased in the early stage of storage and reached a maximum on the seventh day, indicating that beef liver paste showed strong gelation and anti-fluidity at this time. The loss tangent is expressed as the ratio of the loss modulus to the storage modulus [30]. The loss tangent of beef liver paste during storage was less than 1, indicating that it exhibited solid-like properties throughout the storage period.

#### 3.3.5. TBARS and TVB-N

Figure 6 shows the variation of TBARS and TVB-N values during storage for the infant supplement beef liver paste. TBARS is used to indicate the degree of lipids oxidation at the beginning of storage [72]. The TBARS value increased gradually with storage time and reached its maximum value (0.58 mg/kg) on day 7. During storage, the TBARS value was less than 1 mg/Kg and there was no smell of corruption, meaning that the beef liver paste had not deteriorated [73].

TVB-N value of beef liver paste showed an increasing trend with storage time, which could be attributed to the increase in alkaline nitrogenous substances produced by the decomposition of proteins in beef liver paste under the action of enzymes and bacteria [32]. The TVB-N content peaked at 4.98 mg/100g at day 60, and according to the Chinese national standard (GB 2707-2016), TVB-N exceeding 15 mg/100g is considered spoiled. The TVB-N value of the beef liver paste was well below the limit value, indicating that the beef liver paste maintained its fresh quality during storage.

#### 3.3.6. In Vitro Simulated Digestion Evaluation

##### Digestibility

Whether the protein in infant food can be easily digested and absorbed by infants is a key condition for measuring the superiority of protein in infant food. From the results in Figure 7, it can be seen that only a small portion of the protein ingested by infants and young children was digested in the stomach, and most of the protein was digested and absorbed in the small intestine stage. In addition, protein digestibility tends to decrease with increased storage time. This was due to the oxidation of proteins during storage, resulting in the modification of some amino acid side chains, the formation of protein crosslinks, and changes in hydrophobicity, which weakened the binding of proteases to proteins, which in turn led to a decrease in protein digestibility. Similarly, during the storage of fermented turkey sausage, lipid oxidation products induced oxidation of proteins, resulting in decreased digestibility [74].

##### Particle Size

According to Table 7, the DX(10), DX(50) and DX(90) of undigested infant beef liver paste gradually increased during the storage period, reaching a maximum on day 7 due to protein cross-linking and aggregation, which was consistent with the results of previous rheological properties. After that, it gradually decreased with the extension of storage time, which was due to the increase in hydrophobicity caused by protein oxidation during storage, which in turn caused changes in particle size [75]. Moreover, the role of enzymes is also the reason for the reduction in particle size, and the low digestibility of pepsin phase indicates that pepsin does not produce good enzymatic degradation of the protein in beef liver paste. And the internal particles of beef liver paste still aggregate under this acidic condition, and after digestion by pancreatic protease, the particle size was significantly reduced, possibly due to the increase in enzyme contact sites, resulting in protein decomposition into smaller particles [76].

##### SDS-PAGE

In the undigested electropherograms of the infant supplement beef liver paste at each storage period (Figure 8A), relatively clear protein bands were observed at day 0 and more pronounced protein bands at 135 kD at day 7, which may be due to cross-linking and aggregation between proteins resulting in more pronounced protein bands. After gastric digestion, a significant decrease in band strength was observed for some large molecular weight protein bands, with only bands below 35 kDa appearing (Figure 8B). Similar results were observed during refrigeration of vacuum-packed pork [77], which may be due to degradation of large molecule proteins to small molecule proteins or peptides by pepsin. After further intestinal digestion (Figure 8C), the large molecular weight bands almost completely disappeared, probably due to further breakdown of small molecule proteins into smaller peptides or free amino acids by trypsin and chymotrypsin [34,78].

### 3.4. Correlation Analysis and Principal Component Analysis

The Pearson correlation coefficient indicates the degree of linear correlation among different indicators, with absolute values less than 0.3 indicating no correlation, 0.3–0.7 indicating weak correlation, and 0.7–1.0 indicating strong correlation. As seen in Figure 9a, storage time showed significant positive correlations (*p* < 0.05) with hardness, moisture, and *a** and *b** and significant negative correlations (*p* < 0.05) with TBARS and TVB-N, indicating that storage time has a significant effect on the physicochemical properties of beef liver paste. In addition, correlations were found between physicochemical indicators during storage. There was a significant positive correlation between hardness and cohesiveness and gumminess and resilience (*p* < 0.05). pH showed a significant negative correlation with TBARS and TVB-N (*p* < 0.05). pH showed a significant negative correlation with *a** (*p* < 0.05) and a significant positive correlation with *b** (*p* < 0.05). There was a significant positive correlation between digestibility and protein content (*p* < 0.05) and a significant negative correlation with particle size (*p* < 0.05).

Principal component analysis (PCA) was used to evaluate the nutritional quality, physicochemical quality, and digestive characteristics of beef liver paste during storage. Figure 9b shows that the first component (PC1) accounted for 45.8% and the second component (PC2) for 29.3%; 0 and 3 days were distributed in the first quadrant, indicating a positive correlation with both PC1 and PC2. Additionally, 7 and 15 d were distributed in the fourth quadrant, indicating a positive correlation with PC1 and a negative correlation with PC2, and 30 and 45 days were located on the reference line of the scoring plot, indicating that there was no significant difference between these two time points and no significant difference between variables. Digestibility, ash, *a**, TVB-N, and TBARS were negatively correlated with PC1. *L**, *b**, pH, fat, protein, moisture, hardness, elasticity, adhesion, cohesiveness, reversibility, and particle size were positively correlated with PC1. Digestibility, *L**, *b**, pH, fat, protein, moisture, hardness, and elasticity were negatively correlated with PC2. Ash, *a**, TVB-N, TBARS, particle size, adhesiveness, cohesiveness, and reversibility were positively correlated with PC2. At the early stage of storage (0, 3 d), the color, pH, and texture of beef liver paste changed significantly. The decrease in pH led to the analysis of the surface water of beef liver paste, and the paste tissue was gradually loose. In the late storage period (45, 60 d), the TVB-N and TBARS of beef liver paste changed significantly, which was due to the gradual oxidative decomposition of protein and fat, leading to a decline in the quality of the beef liver paste.

### 3.5. Cluster Analysis

To further observe the relationship between storage time and the quality of beef liver paste for infant feeding, cluster analysis was performed on samples with storage periods. The results are shown in Figure 9c, and the clustering of storage time is mainly divided into two categories, cluster I is 0, 3, 7, and 15 d and cluster II is 30, 45, and 60 d. The quality indicators can be divided into three categories: the first is polymerization pH, hardness, springiness, *L**, and *a**, *b**; the second is cohesiveness, gumminess, resilience, TBARS, TVB-N, and moisture; and the third category is ash, fat, protein, gastric digestibility, intestinal digestibility, and particle size. The storage period of 30 days was an important point of quality change in beef liver paste, which is consistent with the results of previous PCA analysis. In the early stages, pH, hardness, springiness, and color were the main indicators of change. The decrease in moisture decreases the surface smoothness and refractive index of the paste, resulting in a decrease in textural properties and *L** values. In later stages, protein and lipid oxidation increased, digestibility decreased, and the quality of beef liver paste decreased.

## 4. Conclusions

Beef liver is rich in protein and is a high-quality complementary food ingredient for infants and young children. This study optimized the formula of beef liver paste and studied the changes in quality and digestive characteristics during storage. The quality of beef liver paste decreased slightly during storage, but it did not reach the level of quality deterioration. According to the results of the in vitro digestion simulation study, digestibility decreased gradually with the increase in storage time. Therefore, storage conditions can be improved according to the dietary characteristics and digestive characteristics of infants and young children, and the digestibility of meat protein can be improved to make it more suitable as a supplementary food for infants and young children.

## Figures and Tables

**Figure 1 foods-12-02689-f001:**
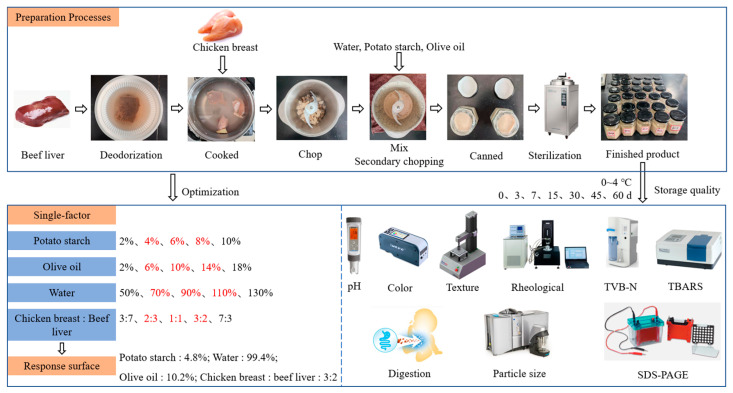
Experimental design for the preparation of beef liver paste as a supplement for infants and children.

**Figure 2 foods-12-02689-f002:**
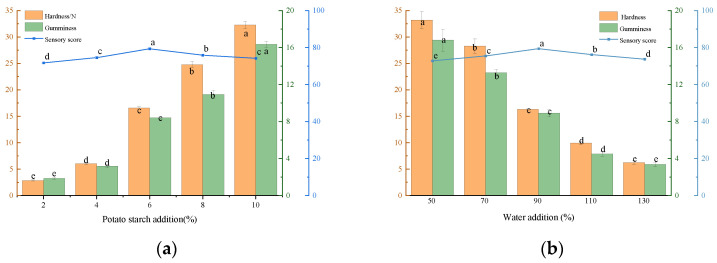
Single-factor optimization of experimental results. (**a**) Effect of adding potato starch to beef liver paste on texture and sensory evaluation. (**b**) Effect of adding water to beef liver paste on texture and sensory evaluation. (**c**) Effect of adding olive oil to beef liver paste on texture and sensory evaluation. (**d**) Effect of adding ratio of chicken breast and beef liver on texture and sensory evaluation of beef liver paste. Different letters (a–e) indicate significant differences (*p* < 0.05).

**Figure 3 foods-12-02689-f003:**
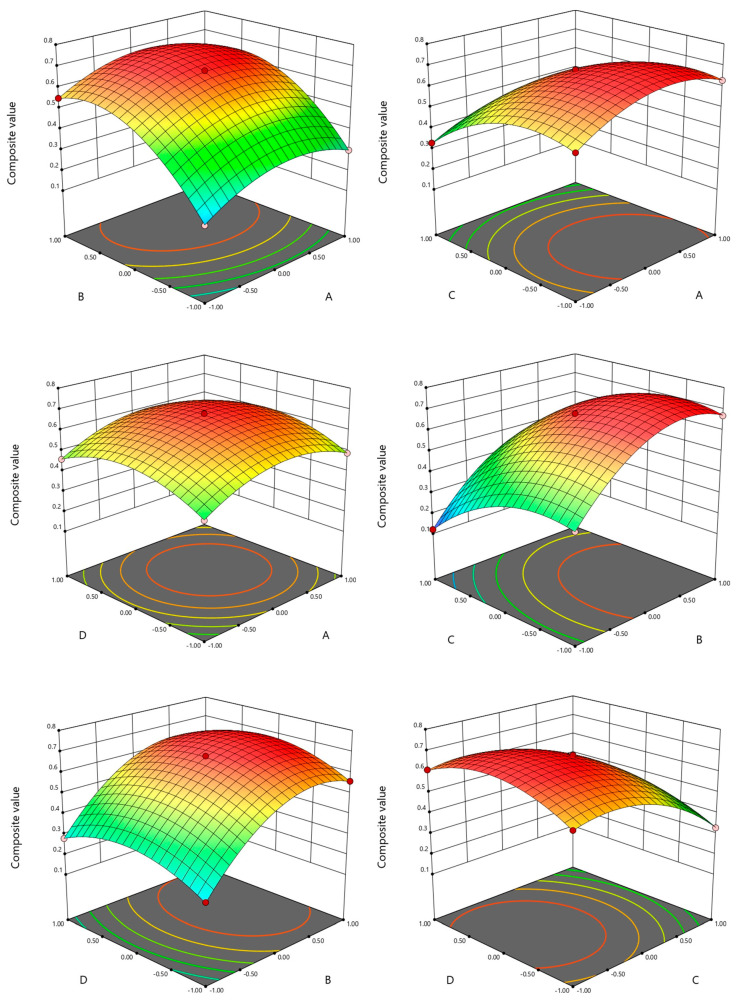
The influence of the interaction of various factors on the comprehensive value of beef liver paste.

**Figure 4 foods-12-02689-f004:**
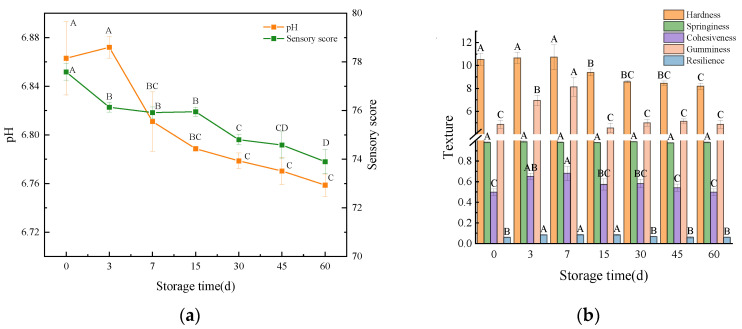
(**a**) Changes in pH of beef liver paste during storage. (**b**) Changes in texture of beef liver paste during storage. Different letters indicate significant differences between storage times (*p* < 0.05).

**Figure 5 foods-12-02689-f005:**
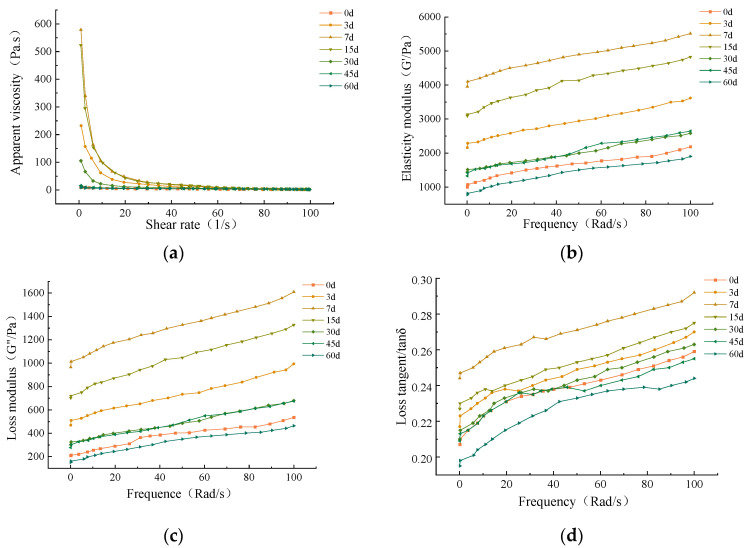
Rheological properties of infant supplementary food beef liver paste during storage. (**a**) Apparent viscosity. (**b**) Elasticity modulus. (**c**) Loss modulus. (**d**) Loss tangent.

**Figure 6 foods-12-02689-f006:**
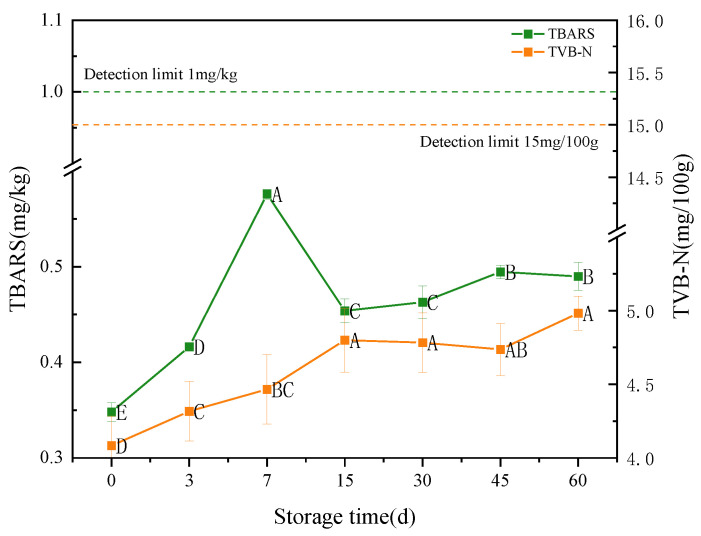
Changes in TBARS values and TVB-N values of infant supplementary food beef liver paste during storage period. Different letters indicate significant differences between storage times (*p* < 0.05).

**Figure 7 foods-12-02689-f007:**
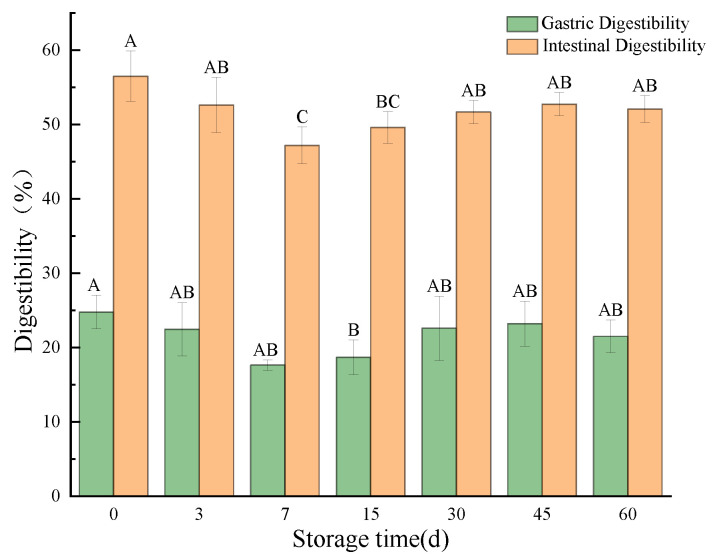
Changes in digestibility of infant supplementary food beef liver paste during storage period. Different letters indicate significant differences between storage times (*p* < 0.05).

**Figure 8 foods-12-02689-f008:**
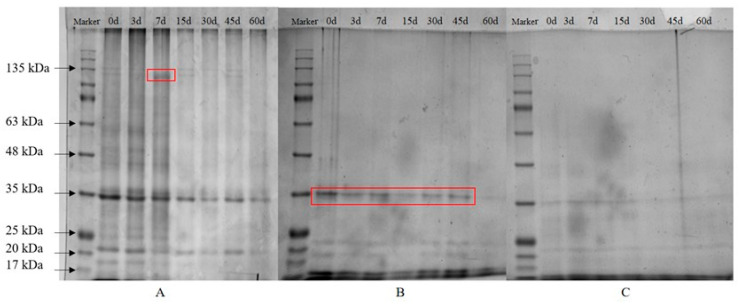
SDS-PAGE of infant supplementary food beef liver paste before and after digestion during storage period. (**A**) Indigestion. (**B**). Gastric digestive products. (**C**) Intestinal digestive products. The red boxed lines mark the places where significant changes occur during storage.

**Figure 9 foods-12-02689-f009:**
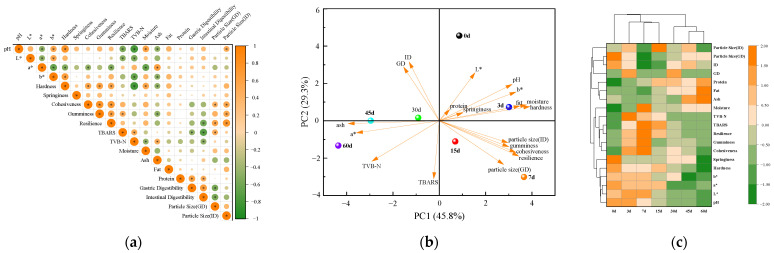
(**a**) Correlation analysis of indicators during the storage of beef liver paste. GD: gastric digestibility; ID: intestinal digestibility. * *p* < 0.05. (**b**) PCA score plots for each component on principal component 1 and principal component 2 during storage of infant supplemental beef liver paste. (**c**) Clustering thermogram of the quality of beef liver paste for infant supplement during storage.

**Table 1 foods-12-02689-t001:** Fuzzy mathematical sensory evaluation criteria.

Items		Excellent (80–100)	Good (60–80)	Medium (40–60)	Poor (10–40)
Taste	Fineness	Fine texture, basically no graininess	Fine taste, slightly grainy	Rough taste, grainy	Coarse taste, heavy graininess
	Softness	Soft taste	Softer taste	Harder taste	Hard taste
	Residue amount	Basically no residue	A small amount of residue	Residue	A lot of residue
Organizational Status	Water-separating property	Basically no water is separated from the tissue	Small amount of water separated from the tissue	Small amount of water separated from the tissue	Water is separated from the tissue and is mobile
	Smearing property	Easy to smear	Relatively easy to smear	Harder to smear	Difficult to smear
Flavor		Liver flavor is rich and basically no fishy taste	Liver flavor is relatively strong and slightly fishy	Liver flavor but not strong and fishy	Liver flavor is not strong and heavy fishy flavor
Color		Shiny and light brown	Slightly shiny and grayish brown	Darker luster and Brown	Basically lusterless and dark brown

**Table 2 foods-12-02689-t002:** Sensory evaluation first level factor weight evaluation results.

Project		Evaluation Team										Total Score
		1	2	3	4	5	6	7	8	9	10	
Primary factor	Taste	6	7	5	9	7	7	9	9	8	7	74
	Organizational status	3	3	4	4	5	3	5	3	6	7	43
	Flavor	7	8	9	6	8	7	6	6	7	4	68
	Color	8	6	6	5	4	7	4	6	3	6	55
	Total	24	24	24	24	24	24	24	24	24	24	240
Secondary factor (Taste)	Fineness	6	4	5	4	6	3	8	8	5	4	43
	Hardness	6	3	2	4	3	6	3	4	6	4	41
	Residue	0	5	5	4	3	3	1	0	1	4	26
	Total	12	12	12	12	12	12	12	12	12	12	120
Secondary factor (Tissue state)	Water-separating property	1	2	1	2	1	1	2	1	3	2	16
	Smearing property	3	2	3	2	3	3	2	3	1	2	24
	Total	4	4	4	4	4	4	4	4	4	4	40

**Table 3 foods-12-02689-t003:** Response surface test design scheme and results.

Run	A	B	C	D	Y1	Y2	Y3	Y
1	−1	0	−1	0	10.02	4.82	74.16	0.53
2	0	0	0	0	16.4	8.33	79.25	0.68
3	−1	1	0	0	12.12	5.48	74.95	0.55
4	0	1	0	−1	14.21	6.62	76.03	0.56
5	0	−1	0	1	19.16	9.85	75.1	0.28
6	0	0	0	0	16.45	7.95	79.25	0.67
7	0	0	−1	1	8.16	4.98	75.14	0.61
8	0	1	1	0	17.65	7.92	76.02	0.46
9	0	1	0	1	10.49	5.93	75.43	0.6
10	0	0	0	0	16.22	8.28	79.16	0.68
11	1	0	−1	0	11.92	6.08	76.09	0.63
12	0	0	0	0	16.8	8.4	79.04	0.67
13	0	1	−1	0	8.34	4.1	75.34	0.67
14	0	−1	1	0	26.51	12.07	75.66	0.12
15	1	1	0	0	14.12	6.94	76.88	0.61
16	0	0	0	0	16.05	8.45	79.16	0.67
17	0	−1	−1	0	16.95	7.95	75.02	0.38
18	1	0	0	−1	18.03	8.64	76.77	0.49
19	1	−1	0	0	22.73	10.83	76.54	0.3
20	−1	0	0	1	12.39	6.51	74.34	0.46
21	0	−1	0	−1	22.54	10.49	75.69	0.25
22	−1	−1	0	0	21.19	9.72	74.72	0.22
23	−1	0	1	0	15.99	7.39	74.16	0.33
24	1	0	0	1	14.42	7.86	76.17	0.53
25	1	0	1	0	21.22	10.12	76.74	0.36
26	0	0	1	1	17.72	8.84	75.29	0.36
27	0	0	−1	−1	12.06	5.75	75.24	0.56
28	−1	0	0	−1	15.98	7.29	74.94	0.42
29	0	0	1	−1	21.42	9.62	75.89	0.33

Note: A means ratio of chicken breast and beef liver added; B means water addition; C means potato starch addition; D means olive oil addition. Y1 means hardness (30%), Y2 means adhesion (30%), Y3 means sensory score (40%), and Y means combined value.

**Table 4 foods-12-02689-t004:** Variance analysis.

Items	S.S.	D.F.	M.S.	F-Value	*p*-Value	Sig.
Model	0.73	14	0.052	3734.77	<0.0001	**
A-Ratio of chicken breast and beef liver	0.014	1	0.014	1005.73	<0.0001	**
B-Water addition	0.3	1	0.3	21,598.29	<0.0001	**
C-Potato starch addition	0.17	1	0.17	12,063.93	<0.0001	**
D-Olive oil addition	4.41 × 10^−3^	1	4.41 × 10^−3^	316.5	<0.0001	**
AB	1.00 × 10^−4^	1	1.00 × 10^−4^	7.18	0.018	*
AC	1.23 × 10^−3^	1	1.23 × 10^−3^	87.95	<0.0001	**
AD	0	1	0	0	1	
BC	6.25 × 10^−4^	1	6.25 × 10^−4^	44.87	<0.0001	**
BD	2.50 × 10^−5^	1	2.50 × 10^−5^	1.79	0.2017	
CD	1.00 × 10^−4^	1	1.00 × 10^−4^	7.18	0.018	*
A^2^	0.065	1	0.065	4688.06	<0.0001	**
B^2^	0.15	1	0.15	11,056.41	<0.0001	**
C^2^	0.081	1	0.081	5798.31	<0.0001	**
D^2^	0.062	1	0.062	4457.35	<0.0001	**
Residual	1.95 × 10^−4^	14	1.39 × 10^−5^			
Lack of fit	7.50 × 10^−5^	10	7.50 × 10^−6^	0.25	0.9657	
Pure error	1.20 × 10^−4^	4	3.00 × 10^−5^			
Cor total	0.73	28				
R^2^	0.9997					
R^2^_adj_	0.9995					

Note: A means ratio of chicken breast and beef liver added; B means water addition; C means potato starch addition; D means olive oil addition. S.S.: sum of squares; D.F.: degrees of freedom; M.S.: mean square.; Sig.: significance. * *p* < 0.05; ** *p* < 0.01.

**Table 5 foods-12-02689-t005:** Chemical composition of beef liver paste.

Items	Average Composition
**Composition (g/100 g)**	
Moisture	78.49 ± 0.04
Ash	1.13 ± 0.07
Protein	10.29 ± 0.41
Fat	5.17 ± 0.12
**Amino Acids (g/100 g)**	
Isoleucine	0.23 ± 0.01
Leucine	0.46 ± 0.02
Threonine	0.12 ± 0.01
Phenylalanine	0.17 ± 0.01
Lysine	0.27 ± 0.02
Valine	0.20 ± 0.02
Histidine	0.13± 0.01
Tryptophan	ND
Aspartic Acid	0.29 ± 0.01
Serine	0.14 ± 0.02
Glutamic acid	3.19 ± 0.06
Cysteine	0.03 ± 0.01
Methionine	0.02 ± 0.01
Tyrosine	0.13 ± 0.01
Arginine	0.25 ± 0.01
**Fatty acids (mg/100 g)**	
C10:0	0.26 ± 0.01
C14:0	1.79 ± 0.01
C15:0	0.26 ± 0.01
C16:0	0.34 ± 0.06
C18:0	92.06 ± 0.57
C20:0	3.89 ± 0.04
C22:0	0.82 ± 0.01
C16:1	3.28 ± 0.44
C20:1	4.36 ± 0.22
C20:5	1.41 ± 0.01
C18:1n9c	412.98 ± 8.54
C18:3n3	17.60 ± 1.02
C18:2n6	137.63 ± 2.06
C20:4n6	22.35 ± 0.12
C22:6ns	2.31 ± 0.04
ΣSFA	99.68 ± 0.24
ΣMUFA	423.48 ± 3.98
ΣPUFA	179.81 ± 2.25
ΣPUFA/ΣSFA	0.55 ± 0.01
Σn-3	3.71 ± 0.02
Σn-6	0.82 ± 0.03
Σn-6/Σn-3	0.22 ± 0.01

Note: ND means not detected.

**Table 6 foods-12-02689-t006:** Changes in the chemical composition and color of beef liver paste during storage.

Items	0 d	3 d	7 d	15 d	30 d	45 d	60 d
Moisture (g/100 g)	79.83 ± 0.99 ^A^	79.85 ± 0.17 ^A^	79.84 ± 0.27 ^A^	79.39 ± 0.20 ^AB^	79.57 ± 0.24 ^AB^	78.69 ± 0.48 ^B^	78.86 ± 0.48 ^B^
Protein (g/100 g)	10.57 ± 0.33 ^A^	10.64 ± 0.55 ^A^	10.50 ± 0.44 ^A^	10.71 ± 0.22 ^A^	10.57 ± 0.45 ^A^	10.64 ± 0.33 ^A^	10.50 ± 0.22 ^A^
Ash (g/100 g)	1.13 ± 0.005 ^B^	1.13 ± 0.036 ^B^	1.11 ± 0.03 ^B^	1.14 ± 0.02 ^AB^	1.15 ± 0.02 ^AB^	1.15 ± 0.02 ^AB^	1.18 ± 0.03 ^A^
Fat (g/100 g)	5.21 ± 0.14 ^A^	5.19 ± 0.14 ^A^	5.19 ± 0.12 ^A^	5.17 ± 0.13 ^A^	5.08 ± 0.15 ^A^	5.07 ± 0.11 ^A^	5.09 ± 0.13 ^A^
*L**	55.13 ± 1.07 ^A^	52.10 ± 0.86 ^B^	52.20 ± 0.70 ^B^	51.83 ± 0.93 ^BC^	52.90 ± 0.62 ^B^	52.70 ± 0.62 ^B^	49.93 ± 1.17 ^C^
*a**	2.60 ± 0.43 ^B^	2.40 ± 0.22 ^B^	2.47 ± 0.12 ^B^	2.67 ± 0.29 ^B^	2.33 ± 0.25 ^B^	3.67 ± 0.34 ^A^	4.17 ± 0.31 ^A^
*b**	24.70 ± 0.29 ^A^	24.17 ± 0.33 ^ABC^	24.23 ± 0.17 ^ABC^	24.37 ± 0.25 ^AB^	23.97 ± 0.25 ^BC^	23.60 ± 0.22 ^CD^	23.13 ± 0.52 ^D^
ΔE	-	3.27 ± 1.61 ^AB^	3.05 ± 1.73 ^AB^	3.39 ± 1.51 ^AB^	2.50 ± 1.39 ^B^	2.92 ± 2.14 ^AB^	5.83 ± 0.24 ^A^

Note: Different letters indicate significant differences between storage times (*p* < 0.05).

**Table 7 foods-12-02689-t007:** Particle size analysis of infant supplementary food beef liver paste before and after digestion during storage period.

Processing Group	Storage Time/d	D3.2	D4.3	DX(10)	DX(50)	DX(90)
Undigested	0 d	14.29 ± 0.93 ^C^	127.87 ± 6.23 ^B^	4.81 ± 0.09 ^B^	128.63 ± 4.84 ^C^	268.57 ± 12.27 ^B^
	3 d	14.83 ± 1.39 ^BC^	132.13 ± 8.01^B^	5.11 ± 0.67 ^B^	140.07 ± 6.45 ^BC^	276.20 ± 16.85 ^B^
	7 d	18.71 ± 0.74 ^A^	161.13 ± 6.58 ^A^	6.02 ± 0.67 ^A^	170.57 ± 15.72 ^A^	315.03 ± 33.31 ^A^
	15 d	17.70 ± 0.71 ^A^	152.47 ± 9.4 ^A^	5.24 ± 0.15 ^B^	166.5 ± 10.0 ^A^	293.77 ± 25.79 ^AB^
	30 d	16.07 ± 0.77 ^B^	139.73 ± 5.33 ^B^	4.77 ± 0.45 ^B^	147.83 ± 5.09 ^B^	280.87 ± 7.89 ^AB^
	45 d	15.09 ± 0.12 ^BC^	138.07 ± 4.27 ^B^	4.82 ± 0.2 ^B^	148.67 ± 3.11 ^B^	276.20 ± 8.61 ^B^
	60 d	14.28 ± 0.51 ^C^	133.51 ± 4.4 ^B^	4.92 ± 0.17 ^B^	133.23 ± 5.81 ^BC^	267.51 ± 10.84 ^B^
Gastric digestion	0 d	9.74 ± 0.91 ^BC^	74.23 ± 2.60 ^D^	3.54 ± 0.8 ^ABC^	27.40 ± 2.54 ^B^	171.03 ± 16.13 ^C^
	3 d	10.72 ± 0.58 ^B^	92.24 ± 10.63 ^B^	2.99 ± 0.16 ^C^	35.44 ± 10.93 ^AB^	183.63 ± 14.48 ^ABC^
	7 d	12.43 ± 0.96^A^	110.89 ± 9.02 ^A^	4.31 ± 0.41 ^A^	41.91 ± 6.74 ^AB^	203.27 ± 16.5 ^A^
	15 d	9.95 ± 0.70 ^BC^	87.02 ± 6.91 ^BCD^	2.69 ± 0.32 ^C^	48.40 ± 9.53 ^A^	195.30 ± 8.61 ^AB^
	30 d	10.04 ± 0.73 ^BC^	88.25 ± 4.71 ^BC^	3.94 ± 0.66 ^AB^	33.26 ± 3.64 ^B^	179.17 ± 6.12 ^BC^
	45 d	9.70 ± 0.69 ^BC^	82.72 ± 8.38 ^BCD^	3.97 ± 0.53 ^AB^	31.18 ± 2.71 ^B^	175.47 ± 11.71 ^BC^
	60 d	9.13 ± 0.29 ^C^	75.59 ± 3.07 ^CD^	3.10 ± 0.14 ^BC^	29.22 ± 4.33 ^B^	173.97 ± 4.59 ^BC^
Intestinal digestion	0 d	2.90 ± 0.42 ^C^	7.00 ± 1.32 ^C^	1.35 ± 0.04 ^C^	3.97 ± 0.98 ^B^	17.53 ± 1.22 ^C^
	3 d	3.80 ± 0.57 ^BC^	8.92 ± 0.81 ^AB^	1.63 ± 0.23 ^BC^	5.02 ± 0.82 ^AB^	23.44 ± 1.63 ^A^
	7 d	4.93 ± 0.84 ^A^	10.49 ± 0.87 ^A^	1.96 ± 0.28 ^A^	6.71 ± 0.32 ^A^	21.02 ± 2.20 ^AB^
	15 d	4.30 ± 0.70 ^AB^	9.29 ± 0.73 ^AB^	1.85 ± 0.19 ^AB^	6.09 ± 0.51 ^A^	20.62 ± 0.78 ^B^
	30 d	4.15 ± 0.56 ^AB^	8.57 ± 1.19 ^BC^	1.78 ± 0.12 ^AB^	6.16 ± 1.40 ^A^	19.45 ± 1.18 ^AB^
	45 d	3.87 ± 0.46 ^ABC^	8.38 ± 0.75 ^BC^	1.82 ± 0.05 ^AB^	5.47 ± 0.84 ^AB^	19.31 ± 1.56 ^AB^
	60 d	4.15 ± 0.20 ^AB^	8.25 ± 1.11 ^BC^	1.67 ± 0.14 ^AB^	5.89 ± 0.95 ^A^	19.27 ± 1.55 ^AB^

Note: Different letters indicate significant differences between storage times (*p* < 0.05).

## Data Availability

The data generated from the study are clearly presented and discussed in the manuscript.

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
