# Peer review of "Preparation of Complementary Food for Infants and Young Children with Beef Liver: Process Optimization and Storage Quality"

_foods, 2023, doi:10.3390/foods12142689_

Round 1

Reviewer 1 Report

Preparation of complementary food for infants and young children with beef liver: Process optimization and storage quality

This manuscript is very interesting and opens door for preparing infant food based on liver in industrial conditions. I have only minor comments.

-          Line 139 “Determination of moisture, fat, ash, and protein content was determined by the Association of Official Analytical Chemist (AOAC).” This association analyzed your samples? There is need to reformulate this sentence e.g. “Determination of moisture, fat, ash, and protein content was determined by standardized method generated by AOAC.”

-          Lines 140 and 142 change “is” to “was”. The text describes what was done.

-     Line 171 Why this sentence starts with “And”? Change it as “Instrument was calibrated…”.

-          Line 178: specify the rheometer type, company name and town.

-          Line 182 and other text and figures are using unusual word “energy”. I prefer to use “elasticity modulus G´.

-          Line 214 and equation (4) use for digestion rate abbreviation DT. Why? I prefer to use “DR”.

-      -Line 394 change title of this part 3.3.1 to “Chemical composition and color” because you have Table 6 with color data there.

-          Table 6 needs to be stretched to the left to avoid number mixing on the second line.

-          Lines 418-428 contain color comments. These lines have to be shifted as part of the 3.3.1 subchapter above or below Table 6.

-          Lines 446-447 this text is not corresponding to figure 5 (a). The sentence “the apparent viscosity increases with increasing shear rate”. It do not correspond the figure 5 (a). I recommend you to choose one characteristic shear rate, predict apparent viscosity valid for this shear rate and prepare new figure 5 (a) as dependence on storage time.

-          Figure 5 (a) has incorrect unit for apparent viscosity! Correct unit is Pa.s.

-          Figure 5 (b) remove Energy storage and input there Elasticity modulus.

-          Line 462 Input there Elasticity modulus.

-          Line 468 “1 mg/kg” is correct.

-          Line 771 remove this line with “1”.

Author Response

We appreciate editor and reviewers for your precious time reviewing our paper. Simultaneously, we would like to thank the reviewers for their feedback on our topic “Preparation of complementary food for infants and young children with beef liver: Process optimization and storage quality”. These valuable comments contribute to further revision and improvement of the paper. We have carefully researched the comments and have made revisions to each comment.

Revised text was shown in red (modifications based on questions from editors and reviewers) and blue font (modifications to grammar and spelling errors in the English language) in the revised manuscript file. The detailed responses to reviewers’ comments are listed below, and the exact line numbers where changes were pointed out in the responses. Attached please find the revised version, which we would like to submit for your kind consideration.

Reviewer 2 Report

- Since chicken livers were also used in the study, the title should be revised.

- Analysis should be done frequently in the first days of the storage period and the reason for the wide gap towards the end should be explained. Analyzes should be made more frequently in animal products with high moisture content, which perish quickly. Dried products are normal though.

-How close is the digestibility rate compared to in vivo. Will the physiological change in children be able to tolerate the difference in digestibility? If so, how much is this rate?

-It would be better if microbiological analysis was done here? Because the decrease in pH of the paste on the 7th and 14th days of storage may have resulted from a fermentation and/or biochemical reaction.

-There has been too much oxidation so that the L value has decreased. No alternative suggestions were given to prevent this oxidation.

-If TBARS values increased, why did the L value decrease? The correlation between must be explained!

-Introduction part should be arranged according to a smooth subject flow.

- The purpose of the study should be detailed.

-The characteristics of the animals used, such as age, gender, etc., should be detailed.

-The peroxide number of the olive oil used should be given. Was the TBARS value initially higher from the liver or the olive oil used?

-The fatty acid composition of olive oil should be given so that it should be explained from which ingredients the differences in the fatty acids of the paste originate.

-The article should be checked from beginning to end, there are some typos, they should be corrected.

improved

Author Response

(The authors gave the same response as above.)

Reviewer 3 Report

In the methodology, section about digestablity, I think there should be a step to remove the digestate from the whole product i.e dialysis or ultrafiltration etc

For all methods the form of instructive way of telling what you have done, is not accepted in scientific articales. Just use past tens for all verbs.

some spelling mistakes and repetetion are found through the manuscript

Author Response

(The authors gave the same response as above.)

Round 2

Reviewer 2 Report

All authors revised accordingbto my comments